# UNIVERSAL SENTENCE REPRESENTATIONS LEARNING WITH CONDITIONAL MASKED LANGUAGE MODEL

## ABSTRACT

This paper presents a novel training method, Conditional Masked Language Modeling (CMLM), to effectively learn sentence representations on large scale unlabeled corpora. CMLM integrates sentence representation learning into MLM training by conditioning on the encoded vectors of adjacent sentences. Our English CMLM model achieves state-of-the-art performance on SentEval (Conneau & Kiela, 2018), even outperforming models learned using (semi-)supervised signals. As a fully unsupervised learning method, CMLM can be conveniently extended to a broad range of languages and domains. We find that a multilingual CMLM model co-trained with bitext retrieval (BR) and natural language inference (NLI) tasks outperforms the previous state-of-the-art multilingual models by a large margin. We explore the same language bias of the learned representations, and propose a principle component based approach to remove the language identifying information from the representation while still retaining sentence semantics.

## 1 INTRODUCTION

Sentence embeddings map sentences into a vector space. The vectors capture rich semantic information that can be used to measure semantic textual similarity (STS) between sentences or train classifiers for a broad range of downstream tasks (Conneau et al., 2017; Subramanian et al., 2018; Logeswaran & Lee, 2018b; Cer et al., 2018; Reimers & Gurevych, 2019; Yang et al., 2019a;d; Giorgi et al., 2020). State-of-the-art models are usually trained on supervised tasks such as natural language inference (Conneau et al., 2017), or with semi-structured data like question-answer pairs (Cer et al., 2018) and translation (Subramanian et al., 2018; Yang et al., 2019a). However, labeled and semi-structured data are difficult and expensive to obtain, making it hard to cover many domains and languages. Conversely, recent efforts to improve language models include the development of masked language model (MLM) pre-training from large scale unlabeled corpora (Devlin et al., 2019; Lan et al., 2020; Liu et al., 2019). While internal MLM model representations are helpful when fine-tuning on downstream tasks, they do not directly produce good sentence representations, without further supervised (Reimers & Gurevych, 2019) or semi-structured (Feng et al., 2020) fine-tuning.

In this paper, we explore an unsupervised approach, called Conditional Masked Language Modeling (CMLM), to effectively learn sentence representations from large scale unlabeled corpora. CMLM integrates sentence representation learning into MLM training by conditioning on sentence level representations produced by adjacent sentences. The model therefore needs to learn effective sentence representations in order to perform good MLM. Since CMLM is fully unsupervised, it can be easily extended to new languages. We explore CMLM for both English and multilingual sentence embeddings for 100+ languages. Our English CMLM model achieves state-of-the-art performance on SentEval (Conneau & Kiela, 2018), even outperforming models learned using (semi-)supervised signals. Moreover, models training on the English Amazon review data using our multilingual vectors exhibit strong multilingual transfer performance on translations of the Amazon review evaluation data to French, German and Japanese, outperforming existing multilingual sentence embedding models by $> 5\%$ for non-English languages and by $> 2\%$ on the original English data.

We further extend the multilingual CMLM to co-training with parallel text (bitext) retrieval task, and finetuning with cross-lingual natural language inference (NLI) data, inspired by the success of prior work on multitask sentence representation learning (Subramanian et al., 2018; Yang et al., 2019a)

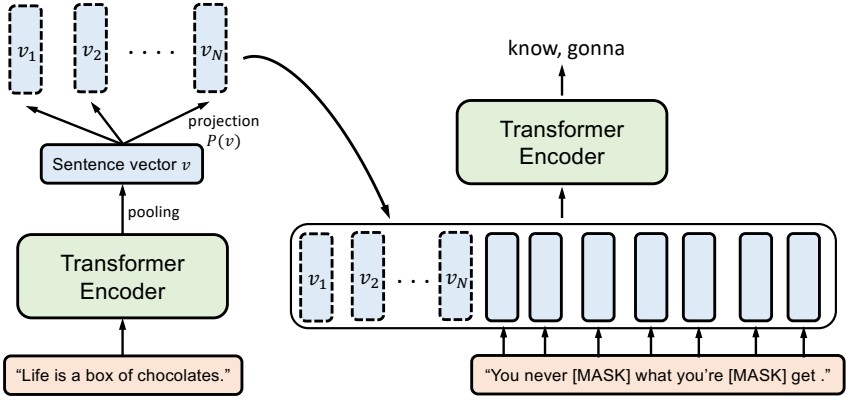

Figure 1: The architecture of Conditional Masked Language Modeling (CMLM).

and NLI learning (Conneau et al., 2017; Reimers & Gurevych, 2019). We achieve performance 1.4% better than the previous state-of-the-art multilingual sentence representation model (see details in section 4.2). Language agnostic representations require semantically similar cross-lingual pairs to be closer in representation space than unrelated same-language pairs (Roy et al., 2020). While we find our original sentence embeddings do have a bias for same language sentences, we discover that removing the first few principal components of the embeddings eliminates the self language bias.

The rest of the paper is organized as follows. Section 2 describes the architecture for CMLM unsupervised learning. In Section 3 we present CMLM trained on English data and evaluation results on SentEval. In Section 3 we apply CMLM to learn sentence multilingual sentence representations. Multitask training strategies on how to effectively combining CMLM, bitext retrieval and cross lingual NLI finetuning are explored. In Section 5, we investigate self language bias in multilingual representations and how to eliminate it.

The contributions of this paper can be summarized as follows: (1) A novel pre-training technique CMLM for unsupervised sentence representation learning on unlabeled corpora (either in monolingual and multilingual). (2) An effective multitask training framework, which combines unsupervised learning task CMLM with supervised learning Bitext Retrieval and cross-lingual NLI finetuning. (3) An evaluation benchmark for multilingual sentence representations. (4) A simple and effective algebraic method to remove same language bias in multilingual representations.

## 2 CONDITIONAL MASKED LANGUAGE MODELING

We introduce Conditional Masked Language Modeling (CMLM) as a novel architecture for combining next sentence prediction with MLM training. By "conditional," we mean the MLM task for one sentence depends on the encoded sentence level representation of the adjacent sentence. This builds on prior work on next sentence prediction that has been widely used for learning sentence level representations (Kiros et al., 2015; Logeswaran & Lee, 2018b; Cer et al., 2018; Yang et al., 2019a), but has thus far produced poor quality sentence embeddings within BERT based models (Reimers & Gurevych, 2019).

While existing MLMs like BERT include next sentence prediction tasks, they do so without any inductive bias to try to encode the meaning of a sentence within a single embedding vector. We introduce a strong inductive bias for learning sentence embeddings by structuring the task as follows. Given a pair of ordered sentences, the first sentence is fed to an encoder that produces a sentence level embedding. The embedding is then provided to an encoder that conditions on the sentence embedding in order to better perform MLM prediction over the second sentence. This is notably similar to Skip-Thought (Kiros et al., 2015), but replaces the generation of the complete second sentence with the MLM denoising objective. It is also similar to T5's MLM inspired unsupervised encode-decoder objective (Raffel et al., 2019), with the second encoder acting as a sort of decoder given the representation produced for the first sentence. Our method critically differs from T5's in that a sentence embedding bottleneck is used to pass information between two model components

and in that the task involves denoising a second sentence when conditioning on the first rather than denoising a single text stream.

Fig. 1 illustrates the architecture of our model. The first sentence $s_1$ is tokenized and input to a transformer encoder and a sentence vector $\boldsymbol{v} \in \mathbb{R}^d$ is computed from the sequence outputs by average pooling.[1] The sentence vector $\boldsymbol{v}$ is then projected into $N$ spaces with one of the projections being the identity mapping, i.e. $\boldsymbol{v}_p = P(\boldsymbol{v}) \in \mathbb{R}^{d \times N}$. Here we use a three-layer MLP as the projection $P(\cdot)$.

The second sentence $s_2$ is then masked following the procedure described in the original BERT paper, including random replacement and the use of unchanged tokens. The second encoder shares the same weights with the encoder used to embed $s_1$ [2]. Tokens in the masked $s_2$ are converted into word vectors and concatenated with $\boldsymbol{v}_p$. The concatenated representations are provided to the transformer encoder to predict the masked tokens in $s_2$. At inference time, we keep the first encoding module and discard the subsequent MLM prediction. In Section 5.2, we explore various different configurations of CMLM, including the number of projection spaces, and how the projected vectors are connected to the embeddings of the second sentence.

## 3 LEARNING ENGLISH SENTENCE REPRESENTATIONS WITH CMLM

For training English sentence encoders with CMLM, we use three Common Crawl dumps. The data are filtered by a classifier which detects whether a sentence belongs to the main content of the web page or not. We use WordPiece tokenization and the vocabulary is the same as public English uncased BERT. In order to enable the model to learn bidirectional information, for two consecutive sequences $s_1$ and $s_2$, we swap their order for $50\%$ of the time. This order-swapping process echos with the preceding and succeeding sentences prediction in Skip-Thought (Kiros et al., 2015). The length of $s_1$ and $s_2$ are set to be 256 tokens (the maximum length). The number of masked tokens in $s_2$ are 80 (31.3%), moderately higher than classical BERT. This change in the ratio of masked tokens is to make the task more challenging, due to the fact that in CMLM, language modeling has access to extra information from adjacent sentences. We train with batch size of 2048 for 1 million steps. The optimizer is LAMB with learning rate of $10^{-3}$, $\beta_1 = 0.9$, $\beta_2 = 0.999$, warm-up in the first 10,000 steps and linear decay afterwards. We explore two transformer configurations, base and large, same as in the original BERT paper. The number of projections $N$ is 15 by experimenting with multiple choices.

### 3.1 EVALUATION

We evaluate the sentence representations on the following tasks: (1) classification: MR (movie reviews Pang & Lee (2005)), binary SST (sentiment analysis, Socher et al. (2013)), TREC (question-type, Voorhees & Tice (2000)), CR (product reviews, Hu & Liu (2004)), SUBJ (subjectivity/objectivity, Pang & Lee (2004)). (2) Entailment: SNLI (Bowman et al., 2015) and SICK dataset for entailment (SICK-E, Marelli et al. (2014)). The evaluation is done using SentEval (Conneau & Kiela, 2018) which is a prevailing evaluation toolkit for sentence embeddings. The classifier for the downstream is logistic regression. For each task, the encoder and embeddings are fixed and only downstream neural structures are trained.

The baseline sentence embedding models include SkipThought (Kiros et al., 2015), InferSent (Conneau et al., 2017), USE (Cer et al., 2018), QuickThought (Logeswaran & Lee, 2018a) and English BERT using standard pre-trained models from TensorFlow Hub website (Devlin et al. (2019), and SBert (Reimers & Gurevych, 2019). To evaluate the possible improvements coming from training data and processes, we train standard BERT models (English BERT base/large (CC)) on the same Common Crawl Corpora that CMLM is trained on. Similarly, we also train QuickThought, a competitive unsupervised sentence representations learning model, on the same Common Crawl Corpora (denoted as "QuickThought (CC)"). To further address the possible advantage from using Transformer encoder, we use a Transformer encoder as the sentence encoder in QuickThought (CC). The representations for BERT are computed by averaging pooling of the sequence outputs

---

[1] One can equivalently choose other pooling methods, such as max pooling or use the vector output corresponding to a special token position such as the [CLS] token.

[2] The dual-encoder sharing encoder weights for different inputs can be also referred as "siamese encoder"

(we also explore options including [CLS] vector and max pooling and the results are available in the appendix).

## 3.2 RESULTS

Evaluation results are presented in Table 1. CMLM outperforms existing models overall, besting MLM (both English BERT and English BERT (CC)) using both base and large configurations. The closest competing model is SBERT, which uses supervised NLI data rather than a purely unsupervised approach. Interestingly, CMLM outperforms SBERT on the SICK-E NLI task.

| Model | MR | CR | SUBJ | MPQA | SST | TREC | MRPC | SICK-E | SICK-R | Avg. |
|---|---|---|---|---|---|---|---|---|---|---|
| SkipThought | 76.5 | 80.1 | 93.6 | 87.1 | 82.0 | 92.2 | 73.0 | 82.3 | 85.8 | 83.8 |
| InferSent | 81.6 | 86.5 | 92.5 | 90.4 | 84.2 | 88.2 | 75.8 | 80.3 | 83.7 | 84.7 |
| USE | 80.1 | 85.2 | 94.0 | 86.7 | 86.4 | 93.2 | 70.1 | 82.4 | 85.9 | 84.9 |
| QuickThought (CC) | 75.7 | 81.9 | 94.3 | 84.7 | 79.7 | 83.0 | 70.4 | 75.0 | 78.5 | 80.4 |
| English BERT base | 81.6 | 87.4 | 95.2 | 87.8 | 85.8 | 90.6 | 71.1 | 79.3 | 80.5 | 84.3 |
| English BERT base (CC) | 82.5 | 88.5 | 95.6 | 87.3 | 88.0 | **91.4** | **72.0** | 79.3 | 79.0 | 84.6 |
| **CMLM base (ours)** | **83.6** | **89.9** | **96.2** | **89.3** | **88.5** | 91.0 | 69.7 | **82.3** | 83.4 | **86.0** |
| English BERT large | 84.3 | 88.9 | 95.7 | 86.8 | 88.9 | 91.4 | 71.8 | 75.7 | 77.0 | 84.5 |
| English BERT large (CC) | 85.4 | 89.0 | 95.7 | 86.9 | 90.5 | 91.2 | 75.5 | 74.3 | 77.0 | 85.0 |
| SBERT (MNLI + SNLI) | 84.8 | **90.0** | 94.5 | **90.3** | 90.7 | 87.4 | **76.0** | 74.9 | 84.2 | 85.9 |
| **CMLM large (ours)** | **85.6** | 89.1 | **96.6** | 89.3 | **91.4** | **92.4** | 70.0 | **82.2** | **84.5** | **86.8** |

Table 1: Transfer learning test set results on SentEval for English models. Baseline models include BERT-based (BERT and SBERT) and non-BERT models (SkipThought, InferSent and USE).

## 4 LEARNING MULTILINGUAL SENTENCE REPRESENTATIONS WITH CMLM

As a fully unsupervised method, CMLM can be conveniently extended to multilingual modeling even for less well resourced languages. Learning good multilingual sentence representations is more challenging than learning monolingual ones, especially when attempting to capture the semantic alignment between different languages. As CMLM does not explicitly address cross-lingual alignment, we explore several modeling approaches besides CMLM: (1) Co-training CMLM with a bitext retrieval task; (2) Fine-tuning with cross-lingual NLI data.

## 4.1 MULTILINGUAL CMLM

We follow the same configuration used to learn English sentence representations with CMLM, but extend the training data to include more languages. Results below will show that CMLM again exhibits competitive performance as a general technique to learn from large scale unlabeled corpora.

## 4.2 MULTITASK TRAINING WITH CMLM AND BITEXT RETRIEVAL

Besides the monolingual pretraining data, we collect a dataset of bilingual translation pairs $\{(s_i, t_i)\}$ using a bitext mining system (Feng et al., 2020). The source sentences $\{s_i\}$ are in English and the target sentences $\{t_i\}$ covers over 100 languages. We build a retrieval task with the translation parallel data, identifying the corresponding translation of the input sentence from candidates in the same batch. Concretely, incorporating Additive Margin Softmax (Yang et al., 2019b), we compute the bitext retrieval loss $\mathcal{L}_{br}^s$ for the source sentences as:

$$\mathcal{L}_{br}^s = -\frac{1}{B}\sum_{i=1}^{B}\frac{e^{\phi(s_i,t_i)-m}}{e^{\phi(s_i,t_i)-m} + \sum_{j=1,j\neq i}^{B} e^{\phi(s_i,t_j)}} \tag{1}$$

Above $\phi(l_s^{(i)}, l_t^{(i)})$ denotes the the inner products of sentence vectors of $l_s^{(i)}$ and $l_t^{(i)}$ (embedded by the transformer encoder); $m$ and $B$ denotes the additive margin and the batch size respectively. Note the way to generate sentence embeddings is the same as in CMLM. We can compute the bitext

retrieval loss for the target sentences $\mathcal{L}_{br}^t$ by normalizing over source sentences, rather than target sentences, in the denominator.[3] The final bitext retrieval loss $\mathcal{L}_{br}$ is given as $\mathcal{L}_{br} = \mathcal{L}_{br}^s + \mathcal{L}_{br}^t$.

There are several ways to incorporate the monolingual CMLM task and bitext retrieval (BR). We explore the following multistage and multitask pretraining strategies:

S1. CMLM+BR: Train with both CMLM and BR from the start;

S2. CMLM → BR: Train with CMLM in the first stage and then train with on BR;

S3. CMLM → CMLM+BR: Train with only CMLM in the first stage and then with both tasks.

When training with both CMLM and BR, the optimization loss is a weighted sum of the language modeling and the retrieval loss $\mathcal{L}_{br}$, i.e. $\mathcal{L} = \mathcal{L}_{CMLM} + \alpha \mathcal{L}_{br}$. We empirically find $\alpha = 0.2$ works well. As shown in Table 3, S3 is found to be the most effective. Unless otherwise denoted, our models trained with CMLM and BR follow S3. We also discover that given a pre-trained transformer encoder, e.g. mBERT, we can improve the quality of sentence representations by finetuning the transformer encoder with CMLM and BR. As shown in Table 2 and Table 3, the improvements between "mBERT" and "f-mBERT" (finetuned mBERT) are significant.

## 4.3 Finetuning with Cross lingual Natural Language Inference

Finetuning with NLI data has proved to be an effective method to improve the quality of embeddings for English models. We extend this to the multilingual domain. Given a premise sentence $u$ and a hypothesis sentence $v$, we train a 3-way classifier on the concatenation of $[u, v, |u - v|, u * v]$. Weights of transformer encoders are also updated in the finetuning process. Different from previous work also using multilingual NLI data (Yang et al., 2019a), the premise $u$ and hypothesis $v$ here are in **different** languages. The cross lingual NLI data are generated by translating Multi-Genre NLI Corpus (Williams et al., 2018) into 14 languages using Google Translate API.

## 4.4 Configurations

Monolingual training data for CMLM are generated from 3 versions of Common Crawl data in 113 languages. The data cleaning and filtering is the same as the English-only ones. A new cased vocabulary is built from the all data sources using the WordPiece vocabulary generation library from Tensorflow Text. The language smoothing exponent from the vocab generation tool is set to 0.3, as the distribution of data size for each language is imbalanced. The final vocabulary size is 501,153. The number of projections $N$ is set to be 15, the batch size $B$ is 2048 and the positive margin is 0.3. For CMLM only pretraining, the number of steps is 2 million. In multitask learning, for S1 and S3, the first stage is of 1.5 million and the second stage is of 1 million steps; for S2, number of training steps is 2 million. The transformer encoder uses the BERT base configuration. Initial learning rate and optimizer chosen are the same as the English models.

## 4.5 Evaluations

### 4.5.1 XEVAL: Multilingual Benchmarks for Sentence Representations Evaluation

Evaluations in previous multilingual literature focused on the cross lingual transfer learning ability from English to other languages. However, this evaluation protocol that treats English as the "anchor" does not equally assess the quality of non-English sentence representations with English ones. In order to address the issue, we prepare a new benchmark for multilingual sentence vectors, XEVAL, by translating SentEval (English) to other 14 languages with an industrial translation API.

Results of models trained with monolingual data are shown in Table 2. Baseline models include mBERT (Devlin et al., 2019), XLM-R (Ruder et al., 2019) and a transformer encoder trained with MLM on the same Common Crawl data (MLM(CC), again this is to control the effects of training data). The method to produce sentence representations for mBERT and XLM-R is chosen to be averaging pooling after exploring options including [CLS] representations and max pooling. The

---

[3]i.e., by swapping the $i$ and $j$ subscripts in the last term of the denominator.

| Model | ar | bg | de | el | en | es | fr | hi | ru | sw | th | tr | ur | vi | zh | Avg. |
|---|---|---|---|---|---|---|---|---|---|---|---|---|---|---|---|---|
| mBERT | 76.3 | 76.1 | 77.7 | 76.1 | 80.1 | 78.5 | 78.7 | 75.6 | 77.3 | 70.5 | 73.6 | 75.7 | 74.2 | 78.8 | 78.7 | 76.5 |
| MLM (CC) | 79.2 | 79.1 | 81.7 | 79.9 | 84.4 | 82.1 | 82.2 | 79.2 | 81.2 | 70.3 | 76.9 | 79.0 | 74.3 | 81.3 | 81.0 | 79.4 |
| XLM-R | 78.1 | 78.0 | 76.2 | 78.2 | 82.8 | 81.2 | 80.4 | 77.2 | 80.2 | 71.0 | 77.5 | 79.7 | 76.7 | 80.3 | 80.8 | 78.5 |
| **CMLM** | **80.6** | **81.2** | **82.6** | **81.4** | **85.0** | **82.3** | **83.4** | **80.0** | **82.3** | **76.2** | **78.8** | **81.0** | **78.5** | **81.6** | **81.7** | **81.2** |

Table 2: Performance (accuracy) of multilingual models trained with monolingual data on XEVAL. Highest numbers are highlighted in bold.

| Model | ar | bg | de | el | en | es | fr | hi | ru | sw | th | tr | ur | vi | zh | Avg. |
|---|---|---|---|---|---|---|---|---|---|---|---|---|---|---|---|---|
| LASER | 82.1 | 81.2 | 81.7 | 78.1 | 82.3 | 81.0 | 80.8 | 78.9 | 82.2 | 75.8 | 80.3 | 81.8 | 77.2 | 81.6 | 82.1 | 80.4 |
| mUSE | 80.4 | 74.0 | 82.2 | 65.0 | 83.3 | 82.7 | 82.4 | 62.3 | 82.3 | 68.1 | 81.6 | 80.3 | 68.8 | 68.0 | 82.0 | 76.2 |
| **S1** | 78.3 | 78.9 | 79.3 | 78.1 | 81.0 | 78.7 | 79.5 | 78.0 | 79.0 | 76.6 | 77.8 | 78.6 | 77.7 | 79.0 | 78.6 | 78.6 |
| **S2** | 81.3 | 81.0 | 83.0 | 81.4 | 85.6 | 83.0 | 83.6 | 80.4 | 82.3 | 77.6 | 80.1 | 81.0 | 79.8 | 82.4 | 82.3 | 81.6 |
| **S3** | 82.6 | 83.0 | 84.0 | 81.8 | 85.8 | 84.2 | 84.6 | 81.7 | 84.0 | 79.3 | 81.2 | 82.7 | 81.2 | 83.0 | 83.0 | 82.8 |
| **S3+NLI** | **84.2** | **83.7** | **85.0** | **83.4** | **87.0** | **85.9** | **85.8** | **83.0** | **85.6** | **79.6** | **83.0** | **84.2** | **81.2** | **84.2** | **84.4** | **84.0** |
| mBERT | 76.3 | 76.1 | 77.7 | 76.1 | 80.1 | 78.5 | 78.7 | 75.6 | 77.3 | 70.5 | 73.6 | 75.7 | 74.2 | 78.8 | 78.7 | 76.5 |
| **f-mBERT** | 77.2 | 78.5 | 79.7 | 76.7 | 81.4 | 80.0 | 80.3 | 77.2 | 79.1 | 73.3 | 76.1 | 77.1 | 76.9 | 79.8 | 80.4 | 78.3 |

Table 3: Performance (accuracy) of models trained with cross lingual data on XEVAL. mUSE only supports 16 languages, underline indicates the language is not supported by mUSE. We test with multiple strategies for multitask pretraining: **[S1]**: CMLM → BR; **[S2]**: CMLM+BR; **[S3]**: CMLM → CMLM+BR. **[f-mBERT]** denotes finetuning mBERT with CMLM and BR.

multilingual model CMLM on monolingual data outperform all baselines in 12 out of 15 languages and the average performance.

Results of models trained with cross lingual data are presented in Table 3. Baseline models for comparison include LASER (Artetxe & Schwenk (2019), trained with parallel data) and multilingual USE ((Yang et al., 2019a), trained with cross lingual NLI). Our model (S3) outperforms LASER in 13 out of 15 languages. Notably, finetuning with NLI in the cross lingual way produces significant improvement (S3 + NLI v.s. S3) and it also outperforms mUSE by significant margins[4]. Multitask learning with CMLM and BR can also be used to increase the performance of pretrained encoders, e.g. mBERT. mBERT trained with CMLM and BR (f-mBERT) has a significant improvement upon mBERT.

### 4.5.2 AMAZON REVIEWS

We also conduct a zero-shot transfer learning evaluation on Amazon reviews dataset (Prettenhofer & Stein, 2010). Following Chidambaram et al. (2019), the original dataset is converted to a classification benchmark by treating reviews with strictly more than 3 stars as positive and negative otherwise. We split 6000 English reviews in the original training set into 90% for training and 10% for development. The two-way classifier, upon the concatenation of $[\boldsymbol{u}, \boldsymbol{v}, |\boldsymbol{u} - \boldsymbol{v}|, \boldsymbol{u} * \boldsymbol{v}]$ (following previous works e.g. Reimers & Gurevych (2019)), is trained on the English training set and then evaluated on English, French, German and Japanese test sets (each has 6000 examples). Note the same multilingual encoder and classifier are used for all the evaluations. We also experiment with whether freezing the encoder weights or not during training. As presented in Table 4, CMLM alone has already outperformed baseline models. Training with BR and cross lingual NLI finetuning further boost the performance.

### 4.6 TATOEBA: SEMANTIC SEARCH

To directly assess the ability of our models on capturing semantics, we test on Tatoeba dataset proposed in Artetxe & Schwenk (2019). Tatoeba dataset include up to 1,000 English-aligned sentence pairs for each evaluated language. The task is to find the nearest neighbor for the query sentence in the other language by cosine similarity. The experiments is conducted on the 36 languages sent as in XTREME benchmark (Hu et al., 2020) and the evaluation metric is retrieval accuracy. We test

---

[4]Note mUSE only supports 16 languages, the best CMLM model is still significantly better if only considering the mUSE supported languages (underline in table 2 indicates the unsupported languages by mUSE)

| Models | English | French | German | Japanese |
|---|---|---|---|---|
| *Encoder parameters are frozen during finetuning* | | | | |
| Eriguchi et al. (2018) | 83.2 | 81.3 | - | - |
| Chidambaram et al. (2019) en-fr | 87.4 | 82.3 | - | - |
| Chidambaram et al. (2019) en-de | 87.1 | - | 81.0 | - |
| mBERT | 80.0 | 73.1 | 70.4 | 71.7 |
| XLM-R | - | 85.3 | 81.5 | 82.5 |
| **CMLM** | 88.4 | 88.2 | 87.5 | **83.7** |
| **CMLM+ BR** | 88.3 | 87.2 | 86.4 | 83.2 |
| **CMLM+ BR + NLI** | **89.4** | **88.8** | **88.4** | 82.8 |
| *Encoder parameters are trained during finetuning* | | | | |
| mBERT | 89.3 | 83.5 | 79.4 | 74.0 |
| **CMLM** | 93.4 | 92.4 | 92.1 | **88.6** |
| **CMLM+ BR** | 93.6 | **93.1** | 92.3 | 88.1 |
| **CMLM+ BR + NLI** | **93.7** | 92.4 | **93.5** | 86.8 |

Table 4: Classification accuracy on the Amazon Reviews dataset. The experiments examine the zero-shot cross-lingual ability of multilingual models. We explore both freezing/updating the weights of the multilingual encoder during training on English data.

our models with configuration CMLM+BR and CMLM+BR+NLI. Baselines (results collected from Hu et al. (2020); Artetxe & Schwenk (2019)) include mBERT, LASER, XLM, XLM-R. Results are presented in Table 5. Our model CMLM+BR outperforms all baseline models in 30 out of 36 languages and has the highest average performance. One interesting observation is that finetuning with NLI actually undermines the model performance on semantic search, in contrary with the significant improvements from CMLM+BR to CMLM+BR+NLI on XEVAL (Table 3). We speculate this is because unlike semantic search, NLI inference is not a linear process. Finetuning with cross-lingual NLI is not expected to help the linear retrieval by nearest neighbor search.

| Lang. | af | ar | bg | bn | de | el | es | et | eu | fa | fi | fr | he | hi | hu | id | it | ja |
|---|---|---|---|---|---|---|---|---|---|---|---|---|---|---|---|---|---|---|
| mBERT | 42.7 | 25.8 | 49.3 | 17 | 77.2 | 29.8 | 68.7 | 29.3 | 25.5 | 46.1 | 39 | 66.3 | 41.9 | 34.8 | 38.7 | 54.6 | 58.4 | 42 |
| XLM | 43.2 | 18.2 | 40 | 13.5 | 66.2 | 25.6 | 58.4 | 24.8 | 17.1 | 32.2 | 32.2 | 54.5 | 32.1 | 26.5 | 30.1 | 45.9 | 56.5 | 40 |
| XLM-R | 58.2 | 47.5 | 71.6 | 43 | 88.8 | 61.8 | 75.7 | 52.2 | 35.8 | 70.5 | 71.6 | 73.7 | 66.4 | 72.2 | 65.4 | 77 | 68.3 | 60.6 |
| LASER | 89.4 | **91.9** | 95.0 | 89.6 | **99.0** | 94.9 | 98.0 | **96.7** | **94.6** | 71.6 | **96.3** | 95.6 | 92.1 | 94.7 | 96.0 | 94.5 | **95.4** | 95.3 |
| **CMLM+BR** | **96.3** | 90.6 | **95.4** | **91.2** | 97.7 | **95.4** | **98.1** | 95.6 | 92.0 | **95.6** | 95.9 | **96.1** | **92.8** | 97.6 | 96.5 | 95.6 | 94.2 | 95.6 |
| **CMLM+BR+NLI** | 90.5 | 83.6 | 92.6 | 86.4 | 97.6 | 91.6 | 9.5 | 82.6 | 76.3 | 90.7 | 88.9 | 93.5 | 86.8 | 94.6 | 89.6 | 91.7 | 90.4 | 88.4 |

| | jv | ka | kk | ko | ml | mr | nl | pt | ru | sw | ta | te | th | tl | tr | ur | vi | zh | Mean |
|---|---|---|---|---|---|---|---|---|---|---|---|---|---|---|---|---|---|---|---|
| mBERT | 17.6 | 20.5 | 27.1 | 38.5 | 19.8 | 20.9 | 68 | 69.9 | 61.2 | 11.5 | 14.3 | 16.2 | 13.7 | 16 | 34.8 | 31.6 | 62 | 71.6 | 38.7 |
| XLM | 22.4 | 22.9 | 17.9 | 25.5 | 20.1 | 13.9 | 59.6 | 63.9 | 44.8 | 12.6 | 20.2 | 12.4 | 31.8 | 14.8 | 26.2 | 18.1 | 47.1 | 42.2 | 32.6 |
| XLM-R | 14.1 | 52.1 | 48.5 | 61.4 | 65.4 | 56.8 | 80.8 | 82.2 | 74.1 | 20.3 | 26.4 | 35.9 | 29.4 | 36.7 | 65.7 | 24.3 | 74.7 | 68.3 | 57.3 |
| LASER | 23.0 | 35.9 | 18.6 | 88.9 | 96.9 | 91.5 | 96.3 | 95.2 | 94.4 | 57.5 | 69.4 | 79.7 | 95.4 | 50.6 | 97.5 | 81.9 | 96.8 | 95.5 | 84.4 |
| **CMLM+BR** | **83.4** | **94.9** | **88.6** | **92.4** | **98.9** | **94.5** | **97.3** | 95.3 | **94.9** | **87.0** | **91.2** | **97.9** | **96.6** | 95.3 | **98.6** | **94.4** | **97.5** | 95.6 | **94.7** |
| **CMLM+BR+NLI** | 66.9 | 88.1 | 80.3 | 85.6 | 94.9 | 90.7 | 93.2 | 92.3 | 91.7 | 76.7 | 88.6 | 92.8 | 94.7 | 82.0 | 94.3 | 84.7 | 94.3 | 93.1 | 88.8 |

Table 5: Tatoeba results (retrieval accuracy) for each language. Our model CMLM+BR achieves the best results on 30 out of 36 languages.

# 5 ANALYSIS

## 5.1 LANGUAGE AGNOSTIC PROPERTIES

**Language Agnosticism** has been a property of great interest for multilingual representations. However, there has not been a **qualitative** measurement or rigid definition for this property. Here we propose that "language agnostic" refers to the property that sentences representations are neutral w.r.t their language information. For example, two sentences with similar semantics should be close in embedding space whether they are of the same languages or not. Another case is that given one query sentence in language $l_1$ and two candidate sentences with the identical meanings (different from the query sentence) in languages $l_1$ and $l_2$, the $l_1$ input sentence should not be biased towards the $l_1$ candidate sentence. To capture this intuition, we convert the PAWS-X dataset (Yang et al., 2019c) to a retrieval task to measure the language agnostic property. Specifically, PAWS-X dataset

consists of English sentences and their translations in other six languages ($x$-axis labels in Fig. 2). Given a query, we inspect the language distribution of the retrieved sentences (by ranking cosine similarities). In Fig. 2, query sentences are in German, French and Chinese for each row. Representations of mBERT (first row) have a strong self language bias, i.e. sentences in the language matching the query are dominant. In contrast, the bias is much weaker in our model trained with CMLM and BR (the third column), probably due to the cross lingual retrieval pretraining. We discover that removing the first principal component of each monolingual space from sentence representations effectively eliminate the self language bias. As shown in the second and the fourth column in Fig. 2, with principal component removal (PCR), the language distribution is much more uniform. We further explore PCR by experimenting on the Tatoeba dataset. Table 5 shows the retrieval accuracy of multilingual model with and w/o PCR. PCR increases the overall retrieval performance for both two models. This suggests **the first principal components in each monolingual space primarily encodes language identification information**.

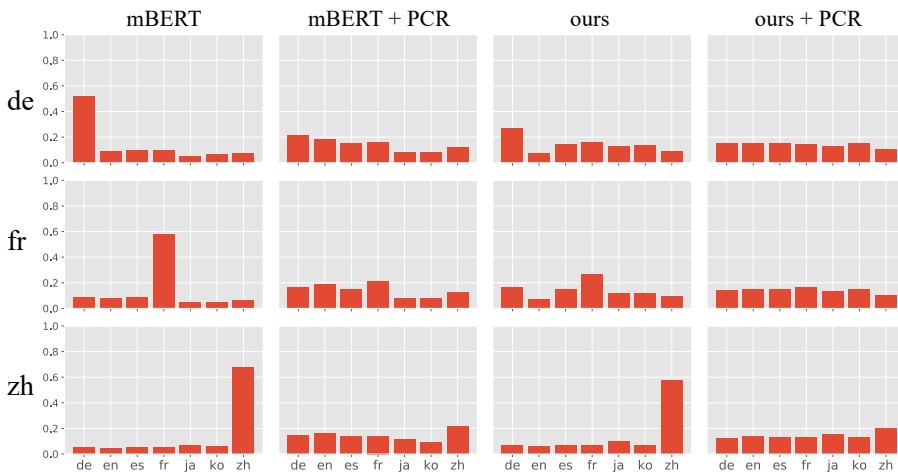

Figure 2: Language distribution of retrieved sentences. The first and third columns are mBERT and our models. Our model already in general has a more uniform distribution than mBERT. The second and fourth columns are mBERT and our model with PCR.

| | fra | cmn | spa | deu | rus | ita | tur | por | hun | jpn | nld | Avg. |
|---|---|---|---|---|---|---|---|---|---|---|---|---|
| mBERT | **60.2** | 60.2 | **62.8** | 65.9 | 53.8 | 55.7 | 32.4 | 62.4 | 31.9 | 39.0 | 56.2 | 52.8 |
| mBERT + PCR | 59.9 | **64.3** | 61.7 | **67.5** | **57.4** | **56.2** | **33.3** | **64.4** | **36.5** | **42.3** | **61.1** | **54.8** |
| ours | **96.1** | 95.6 | 98.1 | 97.7 | 94.9 | **94.2** | **98.6** | 95.3 | 96.5 | **95.6** | **97.3** | 96.3 |
| ours + PCR | 95.5 | **96.0** | **98.2** | **97.9** | **95.1** | 94.1 | 98.5 | **95.8** | **96.6** | 95.3 | 97.2 | **96.4** |

Table 6: Average retrieval accuracy on 36 languages of multilingual representations model with and without principal component removal (PCR) on Tatoeba dataset.

We also visualize the sentence representations in Tatoeba dataset in Fig. 3. Our model (the first row) shows both weak and strong semantic alignment (Roy et al., 2020). Representations are close to others with similar semantics regardless of their languages (strong alignment), especially for French and Russian, where representations form several distinct clusters. Also representations from the same language tend to cluster (weak alignment). While representations from mBERT generally exhibit weak alignment.

## 5.2 ABLATION STUDY

In this section, we explore different configurations of CMLM, including the number of spaces in the projection $N$ and CMLM architecture. As shown in Table 7, projecting the sentence vector into $N = 15$ produces highest overall performance. We also try a modification to CMLM architecture. Besides the concatenation with token embeddings of $s_2$ before input to the transformer encoder, the projected

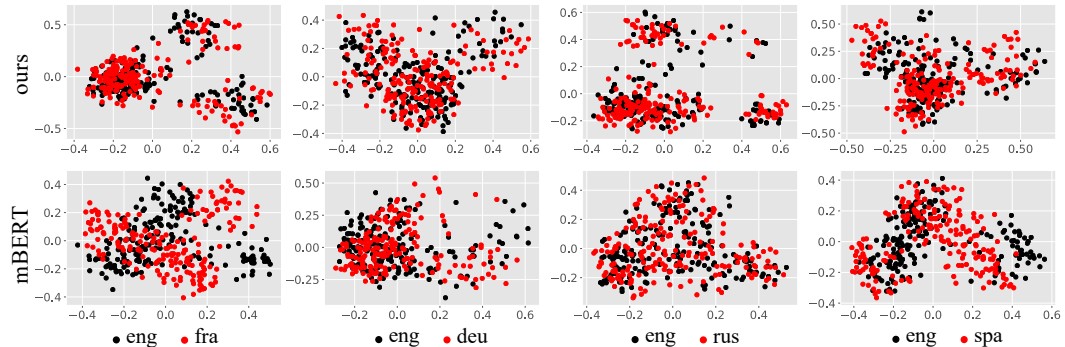

Figure 3: Visualizations of sentence embeddings in Tatoeba dataset in 2D. The target languages are all English and the source languages are French, German, Russian and Spanish from left to right columns. The first and second rows are our model and mBERT respectively.

vectors are also concatenated with the sequence outputs of $s_2$ for the masked token prediction. This version of architecture is denoted as "skip" and model performance actually becomes worse.

Note that the projected vector can also be used to produce the sentence representation $v_s$. For example, one way is to use the average of projected vectors, i.e. $v_s = \frac{1}{N} \sum_i v_p^{(i)}$. Recall $v_p^{(i)}$ is the $i$th projection. This version is denoted as "proj" in Table 7. Sentence representations produced in this way still yield competitive performance, which further confirm the effectiveness of the projection.

| Model | MR | CR | SUBJ | MPQA | SST | TREC | MRPC | SICK-E | SICK-R | Avg. |
|-------|------|------|------|------|------|------|------|--------|--------|------|
| $N = 1$ | 82.3 | 89.7 | 95.8 | 88.8 | 87.6 | 90.4 | **71.5** | 80.8 | 83.4 | 85.5 |
| $N = 5$ | **83.7** | **90.0** | 95.5 | 89.0 | **89.4** | 86.6 | 69.5 | 79.3 | 81.7 | 85.0 |
| $N = 10$ | 83.4 | 89.0 | 96.1 | 88.9 | 88.2 | 90.2 | 68.5 | 79.7 | 81.5 | 84.9 |
| $N = 15$ | 83.6 | 89.9 | **96.2** | **89.3** | 88.5 | **91.0** | 69.7 | **82.3** | **83.4** | **86.0** |
| $N = 20$ | 81.1 | 89.5 | 95.8 | 88.9 | 85.9 | 89.8 | 69.7 | 80.2 | 85.0 | 85.1 |
| skip | 80.3 | 86.8 | 94.5 | 87.5 | 84.9 | 86.0 | 69.2 | 72.8 | 74.7 | 81.9 |
| proj | 82.6 | 89.7 | 96.0 | 87.3 | 87.5 | 89.2 | 70.5 | 81.7 | 83.8 | 85.4 |

Table 7: Ablation study of CMLM designs, including the number of projection spaces, architecture and sentence representations. The experiments are conducted on SentEval.

## 6 CONCLUSION

We present a novel sentence representation learning method Conditional Masked Language Modeling (CMLM) for training on large scale unlabeled corpus. CMLM outperforms the previous state-of-the-art English sentence embeddings models, including those trained with (semi-)supervised signals. For multilingual representations learning, we discover that co-training CMLM with bitext retrieval and cross-lingual NLI finetuning achieves state-of-the-art performance. We also discover that multilingual representations have the same language bias and principal component removal (PCR) can eliminate the bias by separating language identity information from semantics.

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
