# OpenReview forum: "Universal Sentence Representations Learning with Conditional Masked Language Model"
_ICLR.cc/2021/Conference — Reject_

### Official Review · AnonReviewer1 · 2020-10-27
**Low significance, issues in experiments and setup**

**Rating:** 5
**Confidence:** 4

**Review:**

----------

I appreciate the response from authors and the additional experiments. I do think the semantic search task adds value to the paper. However, the paper continues to be centered around the SentEval benchmark results. While SentEval is a useful benchmark to evaluate sentence representations, it doesn't reflect well how these representations will be used in practice. A fine-tuned BERT model will likely perform strongly on these tasks. The paper would be far more compelling if the authors can provide strong evidence that the sentence embeddings do well on tasks where using a BERT model is either less effective due to performance or computational reasons.

I prefer to keep my score.

------------

This work proposes a self-supervised training objective called CMLM (conditional masked language model) for learning sentence representations. An encoder produces multiple fixed length representations of a given sentence and a decoder reconstructs the adjacent sentence given it’s masked version and the encoded representations. CMLM performs well on the SentEval benchmark. CMLM is further extended to the multilingual setting via bi-text retrieval contrastive training and training on NLI data. The multilingual version is shown to work well on multiple translated versions of the SentEval benchmark (SentEval data translated into other languages using an off the shelf translation system) and Amazon reviews (sentiment classification).


Pros
* This work addresses the important problem of (unsupervised) sentence representation learning. Extracting fixed-length sentence representations from popular language model based encoders is a non-trivial problem and this work attempts to provide a solution.
* Experiments go beyond the standard English setting and evaluate sentence representations in the multilingual setting as well.
* Interesting modeling approaches.

Cons
* Experiments are weak. It is unclear to what extent the tasks + evaluation protocol considered here are reflective of language understanding. I don’t think strong baselines were considered. Some of the evaluation benchmarks considered seem arbitrary.
* Model is largely based on prior work. The main contribution is not clear. There are many settings considered in the paper and it is unclear if the proposed contributions are truly significant due to weak baselines are differences in data used for training different methods.

There are several issues with the experimental setup.
* Evaluation protocol: It is unclear if the evaluation protocol considered is measuring language understanding capability well. Representations from the encoders are held fixed and linear classifiers are trained on top of these fixed representations on downstream tasks using labelled data. To me, this is not a setting that demands sentence vectors. It only shows that the sentence vectors capture useful features. I would suggest focusing on a setting where the advantage of the sentence vectors can be demonstrated such as a retrieval problem.
* Baselines: It is unfair to compare the proposed method against baselines like BERT which are not designed to produce fixed length encodings.
* Data used for pre-training: It is difficult to gauge how good the method is in comparison to other models due to differences in the data used for pre-training. Ideally, there should be a table comparing models that use the exact same resources. In Table 1, although BERT-base/large is trained on the same data, it is not a strong baseline since mean pooled representations from the encoder are treated as a sentence representation. Ideally, the model should be compared against a skip-thought baseline or an unsupervised sentence representation learning method that uses the same resources for training.

I would have expected the authors to evaluate multilingual representations on existing benchmarks as well. I don’t find the proposed benchmark XEVAL very convincing. Claims would have been stronger if authors had also included results on existing benchmarks.

The authors need to make it clear exactly what resources are used for training each method.

Presentation can be improved, especially the organization of the paper. It is difficult to follow the paper and identify the main contributions in the current presentation.

The paper touches upon several things - conditional masked language modeling, bitext retrieval, NLI training, language agnosticism, etc, and I find the paper quite incoherent. I suggest the authors make a focused contribution and provide strong experimental evidence to support that contribution. Right now there’s too many things which makes it hard to make sense of the paper as whole.

While the approaches considered in the paper have some merit, the significance of this work is unclear due to issues in the evaluation.

---

> ### Author Response · Authors · 2020-11-22
> **Response to Reviewer 1**
>
> Thanks for your review and valuable feedback! Please find our responses below:
>
> **1. About "experiments are weak".**
>
> The evaluation benchmarks in our paper are widely adopted in previous text representation learning works. For example, SentEval [1,2,3], Amazon Reviews [4,5,6] and Tatoeba [7, 8]. The benchmarks i the paper thoroughly examine the capability of a representation system, including semantic alignment (Tatoeba), transfer learning to various kinds of downstream tasks (sentiment analysis (SST, MR, CR), semantic classification, NLI (SICK-E), text similarity (SICK-R) and paraphrase detection (MRPC) in SentEval, XEVAL) and zero-shot cross-lingual transfer learning (Amazon Reviews). Our models consistently show strong performance across these benchmarks. We believe our evaluations are detailed and in-depth.
>
> **2. About "Model is largely based on prior work..."**
>
> We provide a summary of main contributions of this paper in the last paragraph in the introduction section. To reiterate, we propose an unsupervised sentence representation learning method CMLM. To the best of our knowledge, CMLM is a novel model architecture proposed for the first time.
>
> **3. About "Weak Baselines".**
>
> Baseline models considered in the paper include many SOTA sentence representation models: SkipThought, QuickThought, InferSent, USE and LASER. As shown in Table 1-5, CMLM consistently outperform baseline models. To address the effects from differences in training data, we train multiple baselines with the same training data of CMLM, e.g. QuickThought (CC), English BERT large/base (CC). As shown in table 1, CMLM outperform these baselines that are  trained with the same data resources.
>
> **4. About "Evaluation protocol".**
>
> Following your suggestion, we evaluate our models on Tatoeba [7, 8], a multilingual retrieval benchmark, as shown in Table 5. Our models “CMLM+BR” outperforms all baseline models by a significant margin in terms of the average performance. It has the highest accuracy in 30 out of 36 languages.
>
> **5. About "Data used for pretraining: It is difficult to ..."**
>
> Following your suggestion, we train QuickThought, an unsupervised sentence representation learning method, using the same Common Crawl dumps that our models are trained on. To address the possible advantage coming from the Transformer, we use a transformer encoder in QuickThought instead of a GRU (RNN) in the original QuickThought implementation. The model is denoted as QuickThought (CC) in Table 1. Using a transformer encoder and Common Crawl does not make QuickThought better than our model. Also notice that the model XLM-R also uses Common Crawl corpora. Results in Table 2 and 4 shows that our model CMLM still outperforms XLM-R.
>
> **6. About "paper presentation and organization".**
>
> Thanks for the suggestion! We’ve edited the paper to make the story more coherent following your suggestion. Especially, we added a paragraph in the introduction section (the second last one) to describe how the paper is organized.
>
> **7. Extra evaluations for multilingual representations on existing benchmarks.**
>
> This is a good idea! As mentioned above, we’ve evaluated on the Tatoeba dataset (table 5). Besides XEVAL, the multilingual representations are also evaluated on Amazon Reviews. On both Amazon Reviews and Tatoeba, our models outperform all baseline models.
>
> **8. About "What resources are used to train each method".**
> + For English and Multilingual CMLM, training data (sec. 3 and sec. 4.1) are generated from three Common Crawl dumps (2020-05, 2020-10, 2020-16, see https://commoncrawl.org/the-data/get-started/). English CMLM takes ~5 days using 64 Cloud TPUs (128 TPU chips total). Training Multilingual CMLM takes ~12 days using 64 Cloud TPUs.
> + Multitask co-training CMLM+BR takes ~5 days 64 Cloud TPUs. Information about BR training data can be found at sec. 4.2.
> + Cross-lingual NLI finetuning takes ~12 hours using 8 cloud TPUs. Information about data used for cross-lingual finetuning can be found at sec. 4.3.
>
> References:
>
> [1] Cer, et al. "Universal sentence encoder." arXiv preprint arXiv:1803.11175 (2018).
>
> [2] Conneau, Alexis, et al. "Supervised Learning of Universal Sentence Representations from Natural Language Inference Data." EMNLP 2017.
>
> [3] Yang, et al. . "Parameter-free Sentence Embedding via Orthogonal Basis." EMNLP-IJCNLP 2019.
>
> [4] Zhou, et al. "Cross-lingual sentiment classification with bilingual document representation learning." ACL 2016.
>
> [5] Chidambaram, et al. "Learning Cross-Lingual Sentence Representations via a Multi-task Dual-Encoder Model." RepL4NLP-2019. 2019.
>
> [6] Xu, et al. "Cross-lingual Distillation for Text Classification." ACL 2017.
>
> [7] Artetxe, et al. "Massively multilingual sentence embeddings for zero-shot cross-lingual transfer and beyond." ACL 2019.
>
> [8] Hu, et al. "Xtreme: A massively multilingual multi-task benchmark for evaluating cross-lingual generalization." ICML 2020.

---

> ### Author Response · Authors · 2020-11-24
> **Response to Reviewer 1’s reply**
>
> We appreciate Reviewer 1’s timely response to our rebuttals. Thank you! Please find below our answers to questions raised Reviewer 1.
>
> **“The paper would be far more compelling if the authors can provide strong evidence that the sentence embeddings do well on tasks where using a BERT model is either less effective due to performance or computational reasons.”**
>
> We actually have provided results on such tasks that Reviewer 1 asks for:
>
> - On Amazon Review Dataset, we did try finetuning BERT (Table 4, “Encoder parameters are trained during finetuning”) with in-domain data. In all 4 languages, our models (CMLM, CMLM+BR, CMLM+BR+NLI) have much better performance than finetuned BERT (e.g., 88.6% v.s. 74.0% classification accuracy on Japanese). Especially note that CMLM without finetuning (row “CMLM” in the first group in Table 4) even outperforms finetuned mBERT (row “mBERT” in the second group). This shows that finetuning is not necessarily the only way to produce good embeddings; a well-pretrained sentence representation like CMLM can generate powerful representations..
>
> - We further evaluate languages in the Tatoeba Dataset (table 5). In all 36 languages, our models outperform BERT by significant margins. Concretely, the average retrieval accuracy of our model is 94.7% v.s. 38.7% of mBERT. We believe these two evaluations reflect the case where “a BERT model is less effective due to performance (than CMLM)”.
>
>
> **“While SentEval is a useful benchmark to evaluate sentence representations, it doesn't reflect well how these representations will be used in practice. A fine-tuned BERT model will likely perform strongly on these tasks.”**
>
> In practice, there are many use cases where sentence representations are needed. For example, pre-encoding sentences for retrieval (text records search). Sentence embeddings are still one of the best choices for clustering, retrieval, and modular use of text representations for downstream tasks.
>
> **From these questions raised by Reviewer 1, we feel like the reviewer may have a concern about the research direction of sentence embeddings. Is sentence representation a research direction worth studying when we already have BERT? Why not just finetune BERT?**
>
> - In cases where supervised data are unavailable and you cannot finetune BERT, e.g., clustering and retrieval, having a general-purpose sentence encoding system is crucial for problems.
>
> - Actually in practice, finetuning BERT can result in a drop in performance. In some in-house applications, we observe that finetuning BERT actually yields worse results, e.g. tasks with a single sentence as input. Maybe this is because finetuning makes BERT forget what it learns in pretraining. But in the BERT original paper, the GLUE tasks that BERT shows obvious advantage are those with pair-wised inputs, where BERT-style fine-grained interactions in the finetuning are at advantage. For tasks with single input, there is no strong evidence that a sentence encoding system cannot perform as well as BERT.
>
> - In many non-NLP fields, sentence representations are pre-fixed input features to other systems, which is the same setting that SentEval holds. For example, in Biology [4] and Social Network Analysis [5, 6]. That’s part of why sentence encodings systems like USE (one of most downloaded pre-trained text modules in Tensorflow Hub) and InferSent are still widely used, even after BERT is introduced.
>
> - “How much information and what information we can encode into one sentence” is still an open-ended research problem in NLP [1]. CMLM and explorations on language-agnosticism in this paper provide some insight to this research question.
>
> Reference:
>
> [1] Conneau, Alexis, et al. "What you can cram into a single $ &!#* vector: Probing sentence embeddings for linguistic properties." ACL, 2018.
>
> [2] Cer, et al. "Universal sentence encoder." arXiv preprint arXiv:1803.11175 (2018).
>
> [3] Conneau, Alexis, et al. "Supervised Learning of Universal Sentence Representations from Natural Language Inference Data." EMNLP 2017.
>
> [4] Chen, Qingyu, Yifan Peng, and Zhiyong Lu. "BioSentVec: creating sentence embeddings for biomedical texts." 2019 IEEE International Conference on Healthcare Informatics (ICHI). IEEE, 2019.
>
> [5] Mishra, Rohan, et al. "SNAP-BATNET: Cascading author profiling and social network graphs for suicide ideation detection on social media." Proceedings of the 2019 Conference of the North American Chapter of the Association for Computational Linguistics: Student Research Workshop. 2019.
>
> [6] Wang, Qiaozhi, et al. "# DontTweetThis: Scoring Private Information in Social Networks." Proceedings on Privacy Enhancing Technologies 2019.4 (2019): 72-92.

---

### Official Review · AnonReviewer3 · 2020-10-29
**A reasonable work, but a bit limited in terms of technical contribution, particularly considering that there is not a good intuition explanation.**

**Rating:** 4
**Confidence:** 4

**Review:**

I appreciate the response from the authors to my review, as well as to others.

My concerns on the intuition are most not solved. Although in this DNN dominating era, we cannot expect the explainability as we had before, I still believe that a solid work should be grounded on a reasonable basis, which could be in a high level, such as BERT and SBERT. Let's refer to the example given in the model architecture. The projection of the sentence vector of "Life is a box of chocolates" is left-concatenated with the masked embeddings of the second sentence. This operation is very much lacking in intuition, how come the projection of a sentence representation can be concatenated with the embeddings? In addition, "The second encoder shares the same weights with the encoder used to embed s1", considering their inputs are very different, weight sharing for the two encoders are also problematic.

Another point I just noticed, although the authors claimed that their model is better than SBERT, and did a comparison with SBERT-large, they did not compare with SBERT-base, which makes the conclusion unreliable.

---------------------------------------------------------------------------------------------------------------------------
This paper proposes a method called "Conditional Masked Language Model" for unsupervised sentence representation learning. The method involves two-sentence encoder, where one sentence depends on the encoded sentence level representation of the adjacent sentence. The experimental results are good overall, as the proposed method tends to give the best results across monolingual and multilingual benchmark datasets.

There are still some concerns about the novelty of this paper.  First, I think the explanations for the intuition of the proposed model can be clarified, especially in the introduction section. Second, the baselines used for comparison are not complete, which makes me concern about the effectiveness of the model. The proposed model is the combination of Skip-Thought (Kiros et al., 2015) and BERT masked LM (Devlin et al., 2019).  Their experimental results show a detailed comparison of BERT but ignore much about the Skip-Thought. Although the authors mentioned the results of the Skip-Thought model on the SentEval benchmark, the encoder used in the Skip-Thought (Kiros et al., 2015) is RNN while the author used the Transformer Encoder.  I would appreciate a better and fair comparison of the Skip-Thought model by using same transformer encoder and same training corpus.

I am not sure why you used the concatenation when you do the masked LM.  Are there any other ways to do that?  It can be more convincing to see some analysis or results here. Additionally, there is another work titled “DeCLUTR: Deep Contrastive Learning for Unsupervised Textual Representations”, which also focuses on unsupervised sentence representation learning.  Although it is from arxiv, it would be nice that the authors can mention this work.

It seems good that the authors performed many experiments over many different datasets across monolingual and multilingual.  The exploration of the same language bias of the learned representations is also very interesting.

To summarize, the paper is a bit limited in terms of technical contribution, particularly considering that there is not a good intuition explanation, but some analysis in this paper looks good.

---

> ### Author Response · Authors · 2020-11-22
> **Response to Reviewer 3**
>
> Thanks for your review and valuable comments! Please find our responses below:
>
> **1. "What is the intuition for the proposed Model?"**
>
> - As mentioned in the second paragraph in section 2, the intuition behind CMLM is "to make the encoder produce an effective sentence level embedding of the first sentence for better MLM prediction of the second sentence". We followed your suggestion by modifying the introduction section, especially the second paragraph, to make the intuition behind CMLM clearer.
> - The intuition for bitext retrieval (BR) is to make the multilingual representation language agnostic, which is confirmed by the outstanding zero-shot cross lingual performance on Amazon Reviews (Table 4) and high cross lingual accuracy on Tatoeba (Table 5).
> - The intuition for cross lingual NLI finetuning is to provide supervised learning signals and improve the quality of sentence representations. This is supported by the significant improvements from NLI finetuning on Amazon Reviews Dataset (Tabel 5, between rows “CMLM+BR” and “CMLM+BR+NLI”) and XEVAL (Table 3, between rows “S3” and row “S3+NLI”).
>
> **2. "CMLM looks similar to SkipThought. The baselines used for comparison are not complete since authors should compare with baselines (e.g. SkipThought) using Transformer."**
>
> Following your suggestion, we implement QuickThought (a more recent and better unsupervised sentence representation learning model than SkipThought) with Transformer and train with our data. It is denoted as “QuickThought (CC)” in Table 1. Leveraging our data or Transformer does not make QuickThought better than our models.
>
> Also note that our model differs from SkipThought in the following aspects:
> + SkipThought relies on an extra decoder network while CMLM only has the encoder.
> + SkipThought predicts the entire sentence while CMLM predicts masked tokens only so the predictions can be done in parallel.
>
> These two differences make CMLM more efficient to train when compared with SkipThought.
>
> **3. "Are there any other ways besides using "concatenation" in CMLM? It can be more convincing to see some analysis or results here.".**
>
> Yes. And analysis and results were already in the paper. We are sorry if results were not presented more explicitly. Concretely, besides the “concatenation”, we also tried the “skip connection” configuration in CMLM. Results using this “skip connection” are presented in Table 7, row “skip”. The model architecture of “skip connection” is as follows. Given two sentences $s_1$ and $s_2$. By inputting $s_2$ to the transformer encoder, we obtain an output $M \in R^{H \times L}$, where H denotes the hidden size and L denotes the maximum token length. Recall the sentence representation of s1 is computed as $v \in R^{H}$. We then concatenate $v$ to each column $m_i$ ($i = 1,2,\dots,L$) in $M$. The concatenated tensor $M'$ is of size $2H\times L$, We then use $M’$ as the input for masked token prediction in $s_2$. As shown in table 2, our current configuration “concatenation” is better.
>
> **4. Citations.**
>
> Thanks for pointing out this paper! We’ve cited the DeCLUTR paper as suggested.
>
> **5. We add extra experiments on Tatoeba Semantic Retrieval Dataset to the paper.**
>
> We further evaluate our models on the Tatoeba dataset, as shown in Table 5. Our models “CMLM+BR” outperforms all baseline models by a significant margin in terms of the average performance. It also has the highest accuracy in 30 out of 36 languages.

---

### Official Review · AnonReviewer4 · 2020-10-29
**SkipThought + MLM: worth exploring, more details would improve the article**

**Rating:** 7
**Confidence:** 4

**Review:**

The authors present conditional masked language modeling (CMLM), a new method for unsupervised pretraining, in which the skip-thought notion of conditioning on neighboring sentences is adopted for masked language modeling. The upshot of the proposed approach is that it generates single sentence embeddings that perform competitively on SentEval. In the multilingual setting, the authors combine their CMLM method with a bitext retrieval objective (selecting a sentence’s translation from the other sentences of the language in the batch) that increases performance on a version of the SentEval tasks translated into 14 other languages. In their analysis, the authors make further claims about multilingual embeddings capturing language ID information in their first principle components, a conclusion somewhat substantiated by their results. The authors provide a small amount of ablation experiments for experimental/model design choices.

The underlying idea is worth pursuing, the execution and description could be improved, people will be interested in the results that are present (but then have questions).


Overall

Why only a subset of SentEval for the English experiments (3.2) but then the full SentEval in the multilingual XEval experiment (4.5.1)?  Especially if you are trying to make a single-sentence encoder but then evaluating on SICK-E instead of SICK-R, which is arguably a more applicable eval set.

Why no XLM-R in the amazon reviews analysis (4.5.2)?

Sec 1.
Fig 1: box of chocolate*s*


Sec 2.
--“conditional”, →  “conditional,”

--no quantitative comparison of using max vs mean pool vs CLS embedding

--first sentence is feed → first sentence is fed

--three-layer neural network → three-layer MLP

--refer to using the same set of encoder weights for different inputs as siamese networks, as done in the sentence-bert paper https://en.wikipedia.org/wiki/Siamese_neural_network

-- v_d is used but not defined


Sec 3.
--Skip-thought originally used a sentence to predict the generation of both the preceding and succeeding sentences. This is functionally equivalent to your flip-flopping the order of the consecutive sequences. I would make the point that these steps are equivalent.  Note that this is also not necessarily making the task “more challenging” (and moreover I am not sure why “more challenging” equates with “better pretraining method for language understanding” -- and an ablation of this step is not included to show that it is in fact necessary and useful).

-- similarly, no analysis of masking rate, nor explanation for why ‘more challenging’ is better.

-- “We explore two transformer configurations, base and large, same as in the original BERT paper.” – fragmented

-- The number of projections N = 15. – fragmented

-- SentBERT → SentenceBERT or SBERT

-- On the specific subset of SentEval tasks you’ve selected, the majority of the performance discrepancy is in the SICK-E  task--otherwise, the overall #’s are rather interchangeable. How does this change if you add in the rest of the SentEval tasks, and why were they omitted? Analysis/exploration for why you get such a performance boost only on SICK-E would also be useful.

-- “the length … set to be 256 tokens”: please clarify whether the “length” refers to the maximum length, or each sentence is a fixed-length chunk consisting of 256 tokens

--typo:  “we also exploring”


Sec 4.
--If your introduced bitext retrieval objective uses batch size, experiments comparing the effect of batch size is necessary.

-- Please specify the value of the margin m being used in the experiments

--Choice of number of projections is also not motivated (and in fact contradicted by the ablation  experiment finding that 15 is better)

-- the motivation and contribution for XEVAL are great-- the explanation of the dataset is lacking. What translation API was used? How was the XEVAL score computed for each language? Is it the full set of SentEval downstream tasks?

-- cite precedent for using the concatenation of u,v, u-v and u*v. (or show its effect via ablation)

-- BR  → CMLM+BR configuration not evaluated

-- choice of different training step #’s in each configuration is not particularly motivated.

-- “after exploring options including [CLS] representations and max pooling.” what was the performance drop?

-- typo: “has a significant upon mBERT”


Sec 5.
--It is not clear that you can make the claim that the first PC *only* encodes language-identification information?

--I assume Figure 3 is 2-dimensional TSNE (needs a cite), which comes with its own set of caveats as a visualization tool. Quantitative clustering analysis such as silhouette coefficient might be more appropriate than a plot.  If Figure 3 is not t-SNE, please specify the meaning of X and Y axes.

-- did not try higher than n=15 projections but claimed it was optimal

-- The description of the “skip” ablation is unclear: please clarify what is meant by “concatenated with the sequence outputs of s2”.

-- typos: “removing the first principal component … effectively eliminate”, “for both two models”, “representations … generally exhibits”

---

> ### Author Response · Authors · 2020-11-22
> **Response to Reviewer 4, Part 2**
>
> **Sec 5**
>
> 1. Our claim in the paper is “the first principal components in each monolingual space **primarily** encodes language information”, not “only encodes”. It does not only encode language identification information because note in Table 6, in most cases PCR yields better retrieval performance. For some languages, PCR makes the retrieval performance drop a bit, which indicates that principal components can still contain semantic information.
>
> 2. Figure 3 is a 2D PCA. The x and y axis is the direction of first and second maximum variation through the data. Using silhouette coefficient is a good idea! We’ll add that in the final version.
>
> 3. Explanation for “Concatenated with the sequence outputs of $s_2$”: Given two sentences $s_1$ and $s_2$. By inputting $s_2$ to the transformer encoder, we obtain an output matrix $M \in R^{H\times L}$, where $H$ denotes the hidden size and $L$ denotes the maximum token length. Recall the sentence representation of $s_1$ is computed as $v$ (of size $H$). We then concatenate $v$ to each column vector $m_i$ ($i = 1,2,\dots,L$) in $M$. The concatenated tensor $M’$ is of size $2H\times L$, We then use $M’$ as the input for masked token prediction in $s_2$.

---

> ### Author Response · Authors · 2020-11-22
> **Response to Reviewer 4, Part 1**
>
> Thank you for the detailed comments and insightful suggestions! Please find our responses below:
>
> **Overall:**
> 1. About SentEval: we’ve added results on SICK-R in Table 1. All the numbers in Table 2 and Table 3 are updated by including SICK-R performance on XEVAL.
>
> 2. We’ve added XLM-R performance on Amazon Reviews (Table 4). We also manage to improve the multilingual sentence representations by applying smoothing of the volume of data per language during pretraining (Table 2,3,4).
>
> 3. To have a more thorough understanding of the sentence representations learnt by our models, we added another experiment using the Tatoeba task from the XTREME benchmark (Hu. etc, 2020) dataset that covers 36 languages, shown in Table 5. Our models “CMLM+BR” outperforms all baseline models by a significant margin in terms of the average performance. It also has the highest accuracy in 30 out of 36 languages.
>
> 4. Thanks for pointing out the typos and SBERT naming suggestions! We’ve corrected them in the paper.
>
> **Sec 1, Sec2:**
>
> 1. The quantitative comparison of using MEAN, MAX and CLS is included in the appendix (Table 8).
>
> 2. About siamese networks: we’ve added a footnote on page 3 regarding the name.
>
> 3. $v_d$ should be $v_p$. We’ve corrected this typo in the revision.
>
> **Sec 3:**
>
> 1. About “flip-flopping” and “challenging”: We’ve removed the description of “more challenging” in the first paragraph of section 2. We also add a sentence in the same paragraph to point out the connection between the “order-swapping” in our model and Skip-Thought predictions.
>
> 2. About masking rates: We add analysis on the ablation study of masking ratios in the appendix (Table 9). By “challenging is better”, we mean though low masking ratios yield higher CMLM accuracy in training, it doesn't produce better sentence representations.
>
> 3. About SentEval “subset selection” and SBERT: We’ve included SICK-R results following your suggestion. We’ve included the SentEval tasks that SBERT presented in their original paper (see section 5 in SBERT paper). Also notice that CMLM is only trained on unlabeled corpora while SBERT also uses supervised NLI data. Even so CMLM achieves competitive results. This indicates that CMLM, as an unsupervised sentence representation learning model, is able to obtain sentence representations as good as (if not better) supervised learning models.
>
> 4. About the length 256: The length refers to the maximum length.
>
> **Sec 4:**
>
> 1. About batch size in BR: In general, larger batch sizes improve performance until we reach ~2048, since each example will see more “mismatched” examples. After 2048, we don’t see obvious improvements in performance from increasing batch size. We’ll add detailed results on this in the final version.
>
> 2. The margin m is set to be 0.3. We’ve added this in the paper.
>
> 3. About the number of projections: We’ve included results of N=20 in Table 7. It shows N=15 has a better overall performance than N=20.
>
> 4. The translation is done by Google Translate. The score for each language in XEVAL is computed as the average of performance on tasks MR, CR, SUBJ, MPQA, SST, TREC, MRPC, SICK-E and SICK-R as in the evaluations for English models.
>
> 5. About using the concatenation of u,v, u-v and u*v: We’ve cited previous works using this method.
>
> 6. About BR → CMLM+BR: The suggested path is a great addition. We’ll include the results in the final version. We choose to experiment with CMLM→ CMLM+BR because we notice that BR converges faster than CMLM, therefore we train CMLM first.
>
> 7. About how training step is determined: The training step is determined by the masked token prediction accuracy (CMLM) and retrieval accuracy (BR) on the validation set.
>
> 8. About [CLS] and max-pooling representations: The performance drop of mBERT and XLM-R using [CLS] and max-pooling is very similar to the trend in Table 8 (in appendix)

---

### Official Review · AnonReviewer2 · 2020-10-31
**The results are good but mainly empirical**

**Rating:** 6
**Confidence:** 4

**Review:**

This paper presents Conditional Masked Language Modeling (CMLM), which integrates sentence representation learning into MLM training by conditioning on the encoded vectors of adjacent sentences. It is shown that the English CMLM model achieves strong performance on SentEval, and outperforms models learned using (semi-)supervised signals. It is also found that a multilingual CMLM model co-trained with bitext retrieval (BR) and natural language inference (NLI) tasks outperforms the previous state-of-the-art multilingual models by a large margin. The paper further proposes a principle component based approach to remove the language identifying information from the representation while still retaining sentence semantics.

-Strengths

Learning sentence representations on large scale unlabeled corpora is an important research problem. This paper presents a heavily empirical study, with a series of experiments to evaluate the proposed sentence representation learning method. Multilingual experiments are conducted, with interesting results on language agnostic.

The proposed method, as shown in Figure 1, is somewhat new.

-Weaknesses

The study is mainly empirical.
The authors should provide more details about the three-layer neural network as the projection P (·).
Another concern is that the contribution of this paper to research community may be weak, if the code is not released and the results are not easily reproduced.

--------update after reading the response-----------

Thanks for the authors' response. Mainly empirical and limited in methodology novelty. So I tend to keep the score.

---

> ### Author Response · Authors · 2020-11-22
> **Response to Reviewer 2**
>
> Thank you for your review and valuable suggestions! Please find our responses below:
>
> 1. The architecture of the 3-layer MLP projection $P$ is as follows. Let h denote the dimension of the input sentence vector (e.g. h = $768$ in BERT base; $h = 1024$ in BERT large). Let FC ($a$, $b$, $c$) denote a fully connected layer with input dimension $a$, output dimension $b$ and nonlinearity function $c$. The three layers are FC($h$, $2h$, ReLU), FC($2h$, $2h$, ReLU), FC($2h$, $h$, None). The information has been added to the appendix.
>
> 2. Code and reproduction: we are working on making the code available publicly. Also we will release pretrained models to the public so that researchers can reproduce the results and leverage the model for their own projects. Links will be posted here once available.
>
> 3. To have a more thorough understanding of the sentence representations learnt by our models, we have included an additional experiment on the Tatoeba task from XTREME benchmark [1] that covers 36 languages, shown in Table 5. Our model “CLM+BR” has the highest average performance and outperforms all baseline models on 30 out of 36 languages.
>
> If you have any other questions, please let us know!
>
> References:
>
> [1] Hu, Junjie, et al. "Xtreme: A massively multilingual multi-task benchmark for evaluating cross-lingual generalization." ICML 2020.

---

### Comment · Area_Chair1 · 2020-11-21
**The discussion stage is open!**

Dear Reviewers:

Thanks for your insightful reviews! Now the discussion stage is open and the authors have posted their responses. We will appreciate that the following things-to-do can be done by Tues, Nov 24.

1 Acknowledge explicitly that you have read the responses.

2 Modify your review if necessary.

3 Communicate with the authors/reviewers/AC by adding/responding to the comments if necessary.

Thanks a lot!

---

### Decision · Program_Chairs · 2021-01-07
**Final Decision**

**Decision:**

Reject

**Comment:**

This paper proposes a Conditional Masked Language Modeling (CMLM) method to enhance the MLM by conditioning on the contextual information.

All of the reviewers think the results are good. However, the reviewers also think the intuition and experiments are not so convincing. The responses and revisions still not satisfy all the reviewers' major concern.